# Power and Empowerment in Transdisciplinary Research: A Negotiated Approach for Peri-Urban Groundwater Problems in the Ganges Delta

Leon M. Hermans[1,2,*], Vishal Narain[3], Remi Kempers[4], Sharlene L. Gomes[1], Poulomi Banerjee[5], Rezaul Hasan[6], Mashfiqus Salehin[6], Shah Alam Khan[6], ATM Zakir Hossain[7], Kazi Faisal Islam[7], Sheikh Nazmul Huda[7], Partha Sarathi Banerjee[8], Binoy Majumder[8], Soma Majumder[8], Wil A.H. Thissen[1]

[1]Faculty of Technology, Policy and Management, Delft University of Technology, Delft, the Netherlands
[2]Land and Water Management Department, IHE Delft Institute of Water Education, Delft, the Netherlands
[3]Management Development Institute, Gurgaon, India
[4]Both ENDS, Amsterdam the Netherlands
[5]SaciWATERs, Hyderabad, India
[6]Institute for Water and Flood Management, Bangladesh University of Engineering and Technology, Dhaka, Bangladesh
[7]Jagrata Juba Shangha, Khulna, Bangladesh
[8]The Researcher, Kolkata, India

*Correspondence to*: Leon M. Hermans (l.m.hermans@tudelft.nl / l.hermans@un-ihe.org)

**Abstract.** The co-creation of knowledge through a process of mutual learning between scientists and societal actors is an important avenue to advance science and resolve complex problems in society. While the value and principles for such transdisciplinary water research have been well established, the power and empowerment dimensions continue to pose a challenge, even more so in international processes that bring together participants from the global north and south. We build on earlier research to combine known phases, activities and principles for transdisciplinary water research with a negotiated approach to stakeholder empowerment. Combining these elements, we unpack the power and empowerment dimension in transdisciplinary research for peri-urban groundwater management in the Ganges Delta. Our case experiences show that a negotiated approach offers a useful and needed complement to existing transdisciplinary guidelines. Based on the results, we identify responses to the power and empowerment challenges, which add to existing strategies for transdisciplinary research. A resulting overarching recommendation is to engage with power and politics more explicitly and to do so already from the inception of transdisciplinary activities, as a key input for problem framing and research agenda-setting.

**Key words**: transdisciplinarity, negotiated approach, stakeholder empowerment, peri-urban, groundwater management, Bangladesh, India

# 1 Introduction

Sustainable groundwater management faces various challenges that lend themselves well for transdisciplinary research, including the challenge of social participation and coordinated approaches between multiple actors such as scientists, government agencies and groundwater users (Barthel et al., 2017). This is also true for groundwater management in peri-urban areas. Peri-urban areas are spaces in transition that connect urban and rural environments and that show characteristics of both (Allen 2003; Mc Gee 1991; Singh and Narain 2020). Here, rapid urbanization often results in an increasing pressure on groundwater resources as a source of water for local livelihoods and households, industrial activities and various urban needs. As dynamic spaces in transition, peri-urban areas feature a large diversity and heterogeneity in actors and interests, combined with institutional overlaps, voids and ambiguities (Allen, 2003; Gomes and Hermans, 2017; Narain and Roth, 2022).

In peri-urban areas, water-dependent livelihoods such as farming and fishing may still abound. Proximity to urban and industrial centres may create a spike in real-estate development, and new actors enter the scene. Migrants from more remote rural areas may be attracted by the proximity of urban centres for employment and opportunity, while urban residents and developers may be attracted by available spaces and land. These actors compete for, or threaten the quality of, existing (ground) water resources, such as larger industrial or agro-industrial users, urban water users, and waste (water) disposal activities (Narain et al., 2013; Gomes, 2019). Increased climatic variability, degrading surface water sources, land use change, coupled with unequal power structures, rules, norms and practices, create pressure on already stressed water resources and lead to uncoordinated overexploitation of groundwater aquifers (Narain et al., 2013; Hasan et al., 2019; Banerjee and Hermans, 2020). These increasing demands and pressures, for different users and purposes, are combined with often limited information and knowledge about the actual state of groundwater quantity and quality (Olago, 2019).

Power differences play a large role in the groundwater management in peri-urban areas. As highlighted by political ecology analyses around water governance, power is a key factor shaping differential access to resources (see Swyengedouw 2009; Bryant and Bailey, 1997). Peri-urban water resources tend to be reappropriated and reallocated, whereby some water users get deprived of access to the resource (Banerjee and Hermans, 2020; Narain and Roth, 2022). The resulting lack of access to groundwater during critical periods affects the livelihood securities of the vulnerable and contributes to the incidence of poverty (Banerjee and Jatav, 2017; Butsch et al., 2021).

These combined features of groundwater management in peri-urban areas result in complex situations that match the classic definition of a "wicked" problem situation, at the juncture where conflicting goals and equity issues meet with knowledge limitations and contested problem formulations (Rittel and Webber, 1973). Such complex or wicked problem situations are typically what transdisciplinary research hopes to engage with. Transdisciplinary research has been on the rise as a process of co-creation of knowledge by science and society to offer solutions for complex problems in human-water systems (e.g. Scholz and Steiner, 2015a, Krueger et al., 2016; Ferguson et al., 2018; Ghodsvali et al. 2019; Sapkota, 2019; Pohl et al., 2021). In this co-production of knowledge, stakeholder participation and empowerment, as well as dealing with institutional ambiguity and

informality, is part and parcel of the effort, albeit a very challenging one (Massuel et al., 2018; Ghodsvali et al., 2019; Van Breda and Swilling, 2019).

In transdisciplinary research, the differences in the types of knowledge and experiences that different groups bring to the table, are mixed with established structures for social interactions and the associated power and political dimensions (Jahn et al., 2012; Krueger et al., 2016; Brown, 2018; Pohl et al., 2021). Who is participating in the joint problem articulation and the research efforts, how are these participants selected and how do they report back to their fellow community members? What is needed for these various groups to effectively communicate with each other, and to appreciate the depth and breadth of each other's knowledge and experience? Especially when dealing with relatively vulnerable communities who are not usually involved in research or decision-making, as is the case for peri-urban communities, these issues of power and empowerment cannot be ignored. Glossing over power inequalities may not always be critical for researchers and the production of new scientific knowledge, but it will not help to resolve wicked problems in ways that are scientifically sound, equitable and socially sustainable.

Thus, there is need for strategies to deal with power differences and empowerment in transdisciplinary water research. In 2013, we started a multi-year transdisciplinary water research project that aimed to support groundwater users in peri-urban communities in Bangladesh and India. An international team of researchers and non-governmental organizations worked to develop new scientific knowledge and approaches, while at the same time develop the capacity of local stakeholders to improve groundwater management. From the start of our activities, we were aware of the need to navigate power differences and of the difficulties of combining meaningful societal activities with scientific research. At the same time, we did not find clear-cut recipes to cope with all those challenges in existing accounts of transdisciplinary research.

In this paper, we reflect on our insights and experiences with transdisciplinary water research and stakeholder empowerment in Bangladesh and India. This is done by complementing insights from the literature on transdisciplinary water research with a negotiated approach for stakeholder empowerment (Leeuwis, 2000; Koudstaal et al., 2011). This negotiated approach accepts that social learning is characterized by power differences and strategic behaviour, rather than presuming a neutral dialogue among equals. It uses principles from negotiation literature to support a transformative change towards more local self-governance of natural resources, while also seeking to use and enhance the joint knowledge base. The next section summarizes the relevant literature on the combination of transdisciplinary research and approaches that help deal with power, empowerment and conflict. This is followed in subsequent sections by case experiences with peri-urban groundwater management in the metropolitan areas of Khulna and Kolkata. The findings from these experiences result in practical lessons and suggestions for a more power-sensitive transdisciplinarity, after which we conclude with some final take-aways.

## 2. Transdisciplinary research and stakeholder capacity development for peri-urban groundwater management

### 2.1 Transdisciplinary research

#### 2.1.1 Core concepts and known challenges in transdisciplinary research

There are various conceptualizations of transdisciplinary research, which describe transdisciplinary research as a process of mutual learning whereby science and society interact to develop new knowledge (Max-Neef, 2005; Jahn et al., 2012; Lang et al., 2012; Brandt et al, 2013; Seidl et al., 2013; Scholz & Steiner 2015a; Brown, 2018; Cundill et al, 2018; Djenontin & Meadow, 2018; Fam et al., 2018). With its emphasis on co-creation of knowledge between scientists and local actors outside academia, it is closely related to, and for many practical purposes often indistinguishable from participatory action research (Whyte et al., 1989; Bradbury, 2015) and other participatory, interactive and community-based approaches (Lang et al., 2012). When it comes to human-water systems, transdisciplinary water research has been explored by Krueger et al., (2016) to see where and how water knowledge is produced in society. Transdisciplinary water research has been used for instance as means for more systemic learning on water security issues (Steelman et al., 2015) and for stakeholder engagement and impact of water scarcity modelling (Ferguson et al., 2018). Transdisciplinary water research has also been studied for its role in food-water-energy nexus research to support the achievement of sustainable development goals (Ghodsvali et al., 2019).

All these approaches use a systematic method of inquiry to assist societal actors in improving their actions for addressing societal problems (Bradbury, 2015), while generating methodological innovations and new empirical and theoretical knowledge related to the problem field (Lang et al., 2012). In this interaction, different actors bring their own perception of reality, thought-styles, roles and practices of communication, whereby scientific knowledge is combined with understanding rooted in deep experience (Max-Neef, 2005; Jahn et al., 2012; Pohl et al., 2021). In this process, three types of actors play a key role: i) Stakeholders such as local water users and other people directly related to the water resource, NGOs or companies; (ii) legitimized decision-makers such as policy advisors, government officials and elected political representatives; and (iii) the science community with scientists from academia, applied research institutes and think-tanks (Seidl et al., 2013; Scholz and Steiner, 2015a).

Transdisciplinary science generally distinguishes three main phases, each of which has various challenges: problem framing; co-creation of solution-oriented knowledge; and re-integration of knowledge with scientific and societal practice (Jahn, et al., 2012; Lang et al., 2012; Brandt et al, 2013; Scholz & Steiner 2015b; Steelman et al. 2015). Table 1 shows an illustrative list of these phases and their challenges, based on Lang et al. (2012) and Steelman et al. (2015).

INSERT TABLE 1 AROUND HERE

### 2.1.2 The role of societal stakeholders in transdisciplinary research

Table 1 shows that many of the key challenges relate to the interactions between the different types of actors and the representation of their interests. It starts from the very first phases, with a potential lack of awareness, ownership and legitimacy. This is also in line with Jahn et al. (2012), Klenk and Meehan (2015) and Pohl et al. (2021), who observe that without further scrutiny, transdisciplinarity easily conceals problems with differences in values, knowledge and power. Ghodsvali et al. (2019) also note the apparent challenges involved in stakeholder engagement that goes beyond instrumental levels, in transdisciplinary water nexus research.

Some noteworthy exceptions are present though. Brown (2018) describes experiences with collective learning to enable local communities to cope with sustainability challenges. Process structure and open learning attitudes are identified as the two critical ingredients for these collective social learning processes (Brown, 2018). Krueger et al. (2016) discuss fairness and competence as two important criteria for participation in transdisciplinary co-production of knowledge. Fairness signals the need for everyone with an interest to participate, and to be recognized as valid voices in the process. Competence emphasizes the use of clear rules and procedures in the participation process (Krueger et al., 2016). Cundill et al. (2018) similarly stress the importance of careful process design, taking into account the influence of legal agreements, power asymmetries and institutional values and cultures.

Thus, a clear process design, fairness and open attitudes are known principles for stakeholder engagement in transdisciplinary water research. However, applying these principles can be difficult. When it comes to complex and wicked societal problems, knowledge, learning, capacity and power are intertwined (Rittel and Webber, 1973; Brown, 2018: 285). This limits and complicates joint problem solving (Jahn et al., 2012; Klenk and Meehan, 2015) and makes open dialogue, participatory modelling and scientific knowledge limited as source of undisputed solutions (Barnaud et al., 2010). The questions, assumptions and scenarios included in scientific studies will need to reflect those of societal stakeholders, making them inherently subjective and suited for some problem framings but not others (Godinez-Madrigal et al., 2020). Therefore, transdisciplinarity requires approaches for collective learning that navigate the dimensions of power and fairness in the interactions within and between the various groups of scientists, government agencies and societal water users.

### 2.2 Power, empowerment and negotiated approaches for the co-production of knowledge

### 2.2.1 Power and empowerment in transdisciplinary research

Transdisciplinary scholarship is not blind to the issues of power and fairness. For instance, it recognizes the need for, and difficulties in, establishing a safe platform for joint learning and discovery (Jahn et al., 2012). It also recognizes the importance of representation of different types of stakeholders, including local water users and community stakeholders (Seidl et al., 2013; Scholz and Steiner, 2015a; Dyer et al., 2014). Transdisciplinary research in an international and developing world context recognizes the importance of dealing with institutional cultures (Cundill et al., 2018), institutional ambiguity and informality (Van Breda and Swilling, 2019). What the transdisciplinary literature does not yet offer, is guidance on how to enable a process

and platform for reflexivity and joint learning in a context of power differences, conflicting interests and institutional diversity, ambiguity and informality.

Current guidance and experience is shared only through fairly abstract phrases such as the need for "mechanisms to support mutual learning" and taking the necessary time (Raymond et al., 2010). However, in many cases participation requires not just taking the effort and time to invite stakeholder representatives and raise their problem awareness, but also requires empowering

and capacitating different types of stakeholders to participate and collaborate effectively (Richards et al., 2004; Krueger et al., 2016: 380).

In a context of power differences and competing interests, transdisciplinarity requires two types of capacity building and empowerment. It is not just the capacity of all actors to participate in the knowledge and learning process on an equal footing, but also the capacity to influence and act more effectively in processes of problem solving for water management. Since

transdisciplinary water research seeks to combine scientific knowledge development with societal problem solving, those two types of empowerment are of equal importance. Truly engaging with this dual empowerment dimension is relatively novel (Massuel et al. 2018; Steelman et al., 2015: 596).

### 2.2.2. A negotiated approach to empowerment and transdisciplinary problem solving

The need to address power dimensions in stakeholder participation has been recognized by development practitioners (e.g.
Bebbington et al., 2006; Sneddon and Fox, 2007; Barnaud et al., 2010). This has led to different approaches, including a negotiated approach, starting from the shortcomings of participation models such as social learning or participatory decision-making to deal with conflict (Leeuwis, 2000). Rather than approaching participation as collective decision-making or knowledge co-development, participation should be approached as negotiation: "If in practice participatory projects emerge as `arenas of struggle', and if stakeholders tend to act strategically, rather than communicatively, then why not base
methodological approaches on these assumptions?" (Leeuwis, 2000: 946). Including more explicit attention for strategic behaviour would also provide better outcomes of negotiation and participation process for disadvantaged groups (Edmunds and Wollenberg, 2001). Building on work in relevant fields such as consensus building (Susskind et al., 1999), network management (De Bruijn and Ten Heuvelhof, 2008) and negotiation analysis Fisher et al. (2011), different tasks for an integrative negotiation process were thus identified (Leeuwis, 2000).

In parallel to, and interaction with, this academic development, civil society organizations had similar observations and experiences, reaching similar conclusions. Their experiences and the academic reflections transpired into practical guidelines for a negotiated approach (Koudstaal and Paranjpye, 2011). The earlier participation and co-production activities such as ensuring access to knowledge development for local platforms, continuous learning, and recognizing community knowledge as well as rigorous and innovative science, remain still important pillars. However, the negotiated approach uses this as part
of a larger aim, which is a transformation of governance, moving towards self-governance of local communities. For this, it follows the tasks proposed by Leeuwis and Van den Ban (2004) and the notion of principled negotiations as described and popularized by Fisher et al. (2011). In principled negotiations, parties focus on their underlying values and interests, rather

than on positions regarding preconceived specific outcomes. This is somewhat similar to the difference in negotiations between "creating actions, designed to build a bigger pie, and claiming actions, designed to obtain a larger share of the pie so created" (Raiffa, 2002: 2).

The negotiated approach offers eight tasks as guidance, and, as can be seen from Table 2, these tasks connect well to the challenges identified for transdisciplinary research. This is especially visible for the transdisciplinary research challenges related to participation, joint ownership and legitimacy of the process and its outcomes.

[INSERT TABLE 2 AROUND HERE]

## 3. Methodology and data

An open question is how to combine these empowerment processes with transdisciplinary knowledge co-production. We investigate this question, using the main phases for transdisciplinary research as described in Table 1, combined with the main tasks for a negotiated approach as provided in Table 2. In the subsequent sections, we share our experiences with combining transdisciplinary research with the negotiated approach to address the challenges in groundwater management in peri-urban villages near Khulna, Bangladesh and near Kolkata, India.

Over the period 2013 to 2019, an international team of researchers and civil society organizations developed and executed the Shifting Grounds project in Khulna, Bangladesh, and Kolkata, India. This project was financed by the Dutch Research Council under its Urbanizing Deltas of the World programme and had an explicit focus on transdisciplinarity, combing scientific research with sustainable development. In the project, team members from Bangladesh, India and the Netherlands cooperated to enhance understanding and build capacity with local stakeholders to support sustainable groundwater management in peri-urban Kolkatta and Khulna. Project partners in India consisted of SaciWATERs, a consortium for interdisciplinary water research with expertise in socio-economics and peri-urban water governance, and The Researcher, a civil society research organization that supported the community and stakeholder engagement activities in Kolkata. In Bangladesh the Institute of Water and Flood Management of Bangladesh University of Engineering and Technology (BUET) in Dhaka brought in specific groundwater research expertise and the non-governmental organization (NGO) Jagrata Juba Shangha (JJS) facilitated activities in Khulna. From the Netherlands, Both ENDS, an international sustainable development NGO, brought in a long-standing experience with the negotiated approach in water management and the Faculty of Technology, Policy and Management of Delft University of Technology (TU Delft) contributed expertise in water policy and institutional analysis.

The description of our experiences in the next sections is based on a large body of documented meetings, workshop reports, project progress and evaluation reports, research publications and a three-day team reflection and writing workshop at the end of the project, in 2018 in Khulna, Bangladesh. Many of the workshop reports and research publications can be accessed via the Shifting Grounds project website (SaciWATERs et al., n.d.). A report of the final team writing workshop is available as

Hermans et al. (2019). Furthermore, an overview of activities related to capacity building for institutional analysis in this
project is contained in the dissertation of Sharlene Gomes (2019).

In the description of our experiences, we follow the main phases, tasks and activities as identified in Tables 1 and 2 above. In doing so, we pay specific attention to the interactions and interfaces between researchers, local communities, and state/government actors. The three main transdisciplinary research phases of problem framing, co-creating knowledge and re-integrating knowledge help to structure our account, together with the eight negotiated approach tasks. However, it is important to note that activities often overlap and that the process always features various iterations, going back-and-forth between phases and activities. It is less of a linear and more of an interative process.

## 4. Case introduction: The Shifting Grounds project and its early project design and problem framing

The Shifting Grounds project was jointly formulated in 2013 through international workshops of researchers in collaboration with government stakeholders and local community representatives. The aim was to combine research, capacity building and development activities to address peri-urban groundwater problems in cities in Bangladesh and India. Khulna and Kolkata were selected as project cities, being both located in the Ganges delta, sharing some key hydrological and geophysical features, but with different institutional contexts. The international project team sought a conscious mix between a research-initiated process and a community-initiated process to enable a balanced effort of co- creation of both scientific knowledge as well as practical solutions.

The project started with the ambition to combine transdisciplinary research and the negotiated approach, given the expected differences in groundwater access, dependence and power within peri-urban communities. The consortium benefited from earlier research cooperation on peri-urban water security between partners in India and Bangladesh, and from extensive experiences of civil society partner Both ENDS with the negotiated approach. The initial project design targeted peri-urban villages near each of the two cities. Site selection criteria included scientific suitability as well as willingness and (basic) abilities of village stakeholders to engage with the project. For the latter, we looked at the existence of a nucleus for self-organization, such as the presence of an active community-based organization or local village committee that had also identified groundwater-related problems as an important issue for village development. The latter was used to ensure a workable fit with the initial problem framing around groundwater issues, which had been decided early on by the core project team members as a key research gap for peri-urban water security in the region.

The project was designed around three distinct research activities, along with community empowerment. Two PhD researchers and one postdoc researcher were engaged: the first two to study physical groundwater systems and local institutions, respectively, and the third to study socio-economic and livelihoods dynamics. Community empowerment focused on capacity building within the peri-urban communities and on strengthening links of community actors with government processes and state actors. The community empowerment was led by civil society partners in the project consortium and was referred to as the negotiated approach process; the research process was led by the research organizations. Both functioned together as one

team, with joint problem formulation and frequent project team meetings. Key policy-makers and local experts were represented in a Project Advisory Group.

This team constellation was purposefully designed to allow civil society organizations to use their experience and expertise in facilitating (sensitive) processes within the community, while enabling researchers to bring in their research expertise and
knowledge. The frequent meetings within the project team helped provide shared understanding on problem framing and process design, as well as a space where different team members could benefit from each others' strengths, expertises and positions within local and national networks. This also brought sometimes tensions, dilemmas and power differerences inside the project team. Through clear arrangements and agreed responsibilities, combined with frequent meetings, we have tried to navigate those.

**5. Experiences in Kolkata, India**

**5.1 Transdisciplinary Research Phase A: Problem framing and team building**

**Negotiated Approach Task 1: Preparing the process and Task 2: Reaching agreement on process design**

*The research-government interface*
For the activities in Kolkata, the project worked with the two distinct systems in place for decision-making processes in the
state of West Bengal: an administrative and a political system. The administrative government system was run from the state level, via districts, to provide important services to the communities. This administrative system had a hierarchical structure, with an important role for the District Magistrate that operated from Kolkata, and the Block Development Officer at the local level.

In the preparation phase of the project, connections with this administrative system were established via contacts with the
formal decisionmakers and state-level water agencies. Representatives of some of these agencies were invited as members of the project advisory committee. To gain access to these state representatives, the personal network of one of the Indian researchers proved to be essential. The research components in the Shifting Grounds project were highlighted, whereby especially the groundwater research and hydrogeological modelling had the interest of the government actors. The physical science, a cross-country study in Bangladesh and India on groundwater, turned out to be the main selling point in initiating the
contacts with the formal government representatives. At the start of the project implementation, this support from different state government officials also made it possible to get support from the District Magistrate in charge of the district in which the project village was located. Given the relatively hierarchical formal institutional setting and large power distance between district and state-level officials and local level stakeholders, this support was essential to undertake activities with government officials and stakeholders at the local block and village levels.
This created a supportive atmosphere, including state-level experts in the project advisory group, but the longer-term ownership at the district and state levels for the Shifting Grounds project remained limited. Although the groundwater problems in the

peri-urban areas were acknowledged as important issues, the project itself was too much focused on one specific local area, with relatively limited resources, to spark a more intensive involvement from the higher levels of administration.

*The research - community interface*

In parallel to the administrative system, there is a political system with elected representatives at various levels. At the community level, local self-governing bodies are the village councils, panchayats, which are the lowest elected official bodies in rural areas in India. Gram panchayats consist of a number of village councils.

In the beginning, the project team had visited various peri-urban villages to select a suitable project site. In this selection
process, we looked for visible signs of groundwater management problems, for willingness of local stakeholders to work with researchers to address these issues, and for the presence of a certain nucleus of self-organization as sign of a certain level of competence within the village community that our project could build on. The peri-urban village that was eventually selected for this project was located alongside a canal of historic importance, south-east of Kolkata. It is part of the East Kolkata Wetlands, a Ramsar site. Recent developments included a growth in aquaculture, profitable with rising demand for fish in
Kolkata and its suburbs, as well as an increased reliance on groundwater for aquaculture and rice paddy fields.

The project team benefited from the existence of a receptive village leadership. Certain members of the local panchayat shared their knowledge and support and actively participated in project activities from early on. Support from informal local community groups was present through a local youth club and various smaller women self-help groups, who were mobilized with the help of a local panchayat member. An initial informal community meeting was facilitated through the involvement of
a youth club, which was asked to bring people from different occupational groups to ensure diversity in participation.

Access to safe drinking water was a critical issue, identified at the first stages of engagement with the village community in 2015. The existence of a private water-bottling plant inside the village was a controversial issue. The bottling plant was set up on purchased village land and had a bore well installed as the source of bottled water supply. Given these investments and operations, the owner of the bottling plant was a locally powerful figure. In the first project community meetings we noticed
two distinct interest groups, divided in a pro- and anti-bottling plant lobby. One group supported the activities at the bottling plant, sometimes because they would benefit from those as water vendors or workers, while another group considered it an illegitimate appropriation of local groundwater resources in the village.

The local water bottling plant proved to be a very sensitive issue, closely linked to the village power structures and politics. Even before any choices on problem framing were made, the ability to continue within the community was threatened by the
sensitivities over the bottling plant. Therefore, as more information on village problems emerged, the project continued with a more specific focus on what was not the most contentious, but the most crucial issue, shared by groups across the village: access to safe drinking water, free from arsenic risks. This choice was informed by village concerns, combined with and later on confirmed by groundwater research information. In later stages, providing visible contributions to help the villagers cope with the arsenic problems, helped us to build confidence with them and their social and political leaders.

Gradually, the project team realized that the village was very much divided on political lines, a common feature of rural society in the state of West Bengal. The water bottling plant was one issue of contention, but not the only one. This put us in a difficult position. Already from the start, we realized the importance of remaining neutral as a project team, avoiding reliance on current political leaders who might represent one political faction only. At the same time, the village leadership and the officially elected local bodies could not be by-passed, in order not to compromise the participation process and the safety of its

participants. As a result, politics and associated legitimacy questions affected the further stages of the project.

*Experiences within the Shifting Grounds project team*

   The researchers of SaciWATERs (Hyderabad) and TU Delft (the Netherlands) had easier access to the state and district level government officials than the local project organization, The Researcher, in Kolkata. SaciWATERs and TU Delft were

recognized as research institutes of (inter)national importance, which enabled them to access the experts and officials at these levels. The local partner in Kolkata, The Researcher, cultivated a good rapport with the local community representatives. At the same time, across the project team, there was a steep early learning curve on the mechanisms and particularities of the negotiated approach. Even if some guidelines were available, these were fairly generic and their application in the specific setting in West Bengal brought its own challenges and questions. During the first two larger project workshops in Kolkata, the

presence of professor Paranjpye, one of the original developers of the negotiated approach for water management in India, proved essential to support the team in the process design for stakeholder empowerment.

## 5.2 Transdisciplinary Research Phase B: Co-creation of solution-oriented knowledge

### Negotiated Approach Task 3: Joint fact-finding and situation analysis

*Groundwater research and access to official data*

The groundwater modelling, a key research component, struggled with the acquisition of regional-level data for the Kolkata site, despite good contacts with key government officials in the State Water Investigation Directorate. The groundwater researcher had to work with a very limited set of regional level data, combined with some local measurements from a field visit. This constrained the modelling and in-depth site-specific knowledge on the local groundwater situation. Nevertheless, the groundwater knowledge that was available suggested that simply demanding more tube wells for local users might not be

advisable, as it would lower the water table of the village.

   The presence of arsenic is a known issue in the Gangetic delta regions in India and Bangladesh since the 1980s and 1990s. For India, estimates were that about 6.5 million people were affected by severe health risks, using groundwater from affected aquifers for human consumption (Hasan, 2016). A review of groundwater data that were available, supported the focus on arsenic mapping and awareness, as likely risks also for this village. The water quality data that were obtained for the

groundwater research indicated the presence of arsenic, which was validated by the Block Development Officer, Gram Panchayat, and Public Health Department Engineer.

*Institutional research on formal regulations and water rights*

Formal institutions provide a key leverage point for sustaining future interventions and improvement in water management.
For these national and state level policies, acts, and ordinances, an institutions brief was prepared by the institutions researcher to support the negotiated approach process. The brief was presented to the community in their own language, Bengali, printed as a brochure with many pictures and illustrations that made it attractive and helpful to understand, also for the illiterate community members. This was useful as a way of imparting knowledge to the community about people's rights to water and the official government acts and departments regulating water in the state. The community had never heard of such rights to
water or water governing acts. Not all of this knowledge could be translated immediately into action, but the knowledge remained an element of awareness and empowerment on community water rights. Being aware of their rights and the official legal acts and ordinances that are recognized by government bureaucrats and administrators, helps communities become more accepted as partner for dialogue.

*Socio-economic research: Synchronizing longer-term research with short-term community needs*

The initial idea was that household survey results would be used to prepare an integrated groundwater security index, which could be shared with the community to help prioritize issues to be tackled in the negotiated approach. An early survey would also have helped to get a better picture of socio-economic heterogeneity and structures. However, a survey could not be started without initial community engagement and support. As this was initiated, the first community meetings already helped to
prioritize local issues and suggested that some of the issues represented in the scientific groundwater security index might not be relevant locally. Based on this, more questions towards water quality and water distribution were included and wastewater irrigation was added – something that was not there in the standard set-up for the index survey.

Peri-urban spaces are zones of transition and great socio-economic heterogeneity (Allen 2003; Singh and Narain 2020), where the socioeconomic dynamics change very rapidly with regard to status and income. In more remote rural areas it is easier to
understand the status of the people as it is more stable. Conducting the household survey gave the project team a better overview of the problems in the village, especially the differential access to water. The socioeconomic status and dynamics became clear only during the survey, when we visited the households more intensely for several months. The survey also gave us the idea that there was a sizable section of population using groundwater for irrigation. This was not raised in the first community meetings, where the village community had predominantly raised its drinking water problems.

The household survey results eventually were available only well into the third year of the project. At this time, the negotiated approach team had already started working on the particular issue of drinking water and arsenic. Still, the socio-economic research did reinforce earlier choices in the process. We came to know that there were over 900 families in the village with only ten available potable water sources. This reinforced the focus on drinking water.

*Discontinuous participation due to village politics and power shifts*

In the spring of 2016, State Assembly elections were held, resulting in political schisms reaching new heights between rival groups. The deep political divides meant that some community members who had earlier been in leadership positions and had been very supportive to our activities in the initial project stages, could no longer play a role in support of the negotiated approach process. These political dynamics meant that the project team had to make continued efforts by bringing the various lose threads together, roping in new persons and assuaging the conflicting interests to the extent possible. After two of the three initial village 'champions' left the stage due to the rising political tensions, we built rapport with the new leadership of the youth club. One re-elected panchayat member who had previously shown support provided a stable factor and enabled us to connect with the community in the subsequent phases of the process and to maintain contacts with the women's self-help groups. We attended several meetings of these self-help groups, urging the participating women to attend also our informal project community meetings. These efforts ensured a good participation of women in the subsequent meetings.

### 5.3 Transdisciplinary Research Phase C: Re-integrating and applying produced knowledge in science and social practice

**Negotiated Approach Task 4: Solutions analysis and Task 5: Forging agreement**

In a project mid-term review meeting in September 2016, community representatives signalled impatience and dissatisfaction with project progress. Their feeling was that, until then, little direct benefits to the community were visible, endangering their willingness to continue the engagement. They requested the project team to do something concrete in the short-term, to gain confidence of the community and continue the process further. From a pure scientific research perspective, this was difficult to respond to. The research activities were nowhere near finalization and actionable results. Also, the three project researchers by then had differentiated between project sites to focus on in their research, based on access to data, progress in the research and capacity building, and power dynamics. Two researchers were focusing relatively more on Khulna and one researcher was focusing relatively more on Kolkata.

As part of a reciprocal transdisciplinary process, the international project team promised to make an effort to mobilize additional resources to address the pressing issue of arsenic contamination of water sources. This was started in the months after the mid-term meeting and brought in new experts, doctors and equipment to enable actions focused on the arsenic contamination of local domestic water sources. A local arsenic awareness and mapping campaign was started, with arsenic testing of various local water sources and a village health camp. For this, national and local experts were engaged, including a local medical college and water laboratory. This helped to get more detailed information on the local prevalence of arsenic in various water sources, and through the local health camps and workshops, villagers could be checked for symptoms and received medical advice, as well as education about locally developed arsenic removal filters.

**Negotiated Approach Task 6: Communication with constituencies and Task 7: Monitoring and continued involvement**

Tackling the arsenic drinking water quality issue in the village was only possible with the consent of the panchayat officials. After the earlier friction, the panchayat officials eventually recognized the importance of our activities, as they invited us to the local book fair organized by three Gram Panchayats in early-2017, to make an audio-visual presentation on the water
security issue before a larger audience of several hundred people.

As drinking water had been the most crucial issue across the political divide, people belonging to both political sides were involved in the arsenic testing and education process in a more indirect and informal way. During the testing of water samples and the door to door campaign on water quality, political allegiance played no role and people from the opposition camp were also involved. In the formal process, in village meetings and workshops, these people from the political opposition camp were
not always involved. When present in less formal community meetings, they were not so vocal due to fear of being identified. In the second part of the process, we organized an arsenic health camp and an arsenic awareness workshop involving local health workers where arsenic-affected tube wells were marked in maps of all the villages under the panchayat, as part of an effort to address the specific concerns regarding arsenic contamination of domestic water supply. In addition to the direct practical health benefits, these maps and the knowledge gained through these health camps and workshops, enabled the
villagers to better discuss their needs and concerns to government representatives and panchayat bodies.

Monitoring the effects of project interventions and proposed solutions after the project timespan was not possible due to the lack of resources for the project team members to visit the village after the finalization of the project. Although unsatisfactory, this had been anticipated in the project design, whereby we tried to be clear to all stakeholders, and careful ourselves, about our exit from the village after project closure. Part of this exit strategy, for instance, was to steer away from the most
controversial issues around the local drinking water bottling plant, in order not to stir up more conflict than we were capable of handling within the time and resources of our project.

**Negotiated Approach Task 8: Strengthening capacity to become and remain equal partners in negotiations**

A key dimension in the equal partnerships emphasised in the negotiated approach, is gender equality. In the informal meetings and the larger more formal community workshops, women were no less vocal than men. One of the key persons to mobilize
the community for us was a local female panchayat member. That she was a woman probably helped the other women in the community to join our programme in good numbers, as well as to speak out. However, if this lady in our project village had not been re-elected again in the panchayat elections that were held later in the project period in May 2018, our effort to involve the women might have been thwarted. This shows that this effect, although visible, also is fragile.

The Government of India is giving much importance to the panchayats and allocates several hundred crores of rupees for water
supply. Among the formal institutions supposed to be in place at this level, is the Village Water and Sanitation Committee (VWSC), looking after the water and sanitation problems of a gram panchayat area and formally chaired by the panchayat pradhan. However, in the panchayat of our project village, the committee mostly remained on paper. So our aim was to make

it function as the sustainability of the negotiated approach process was dependent on the functioning of this village committee that works for the whole gram panchayat, consisting of seven villages. The panchayat pradhan (chairman) gave us permission and during the final project workshop, members of the committee participated and pledged to use the written project reports with the arsenic testing results to improve the situation. At the time of writing, it remained to be seen whether this committee would remain truly active.

## 6. Experiences in Khulna, Bangladesh

### 6.1 Transdisciplinary Research Phase A: Problem framing and team building

**Negotiated Approach Task 1: Preparing the process**

The village for project activities in Khulna was selected based on pre-scoping visits in the first project phase, reviewing different potential villages as project-sites, similar to the process done for Kolkata. The project activities were then initiated with a community workshop in October 2015. Following this workshop, several smaller group meetings were held in the village to further establish dialogue. Through a series of village level meetings and workshops, people learned about the project and its negotiated approach, while the project team learned more about the village, its stakeholder groups and social dynamics. Land use change is a common feature of peri-urban environments. This is accompanied by a rise in the price of land and efforts at occupational diversification (Narain 2009; Narain and Nischal 2007). This dynamic was also visible in the project village. Traditional fish farming and agriculture were on the decline. Some people were selling their agricultural land to land developers and others to migrants.

During the first visits, it was observed that the village road acted as a rough division between the groups of migrants and permanent residents. The permanent residents were located mostly on the right side of the road, and appeared to be more homogenous, with less rivalling groups within them. The part of the village on the left side of the road had more migrants, who were not as well organized. This made it easier to start the community engagement process mainly at the right side of the road. This, of course, had implications for village representation in the remainder of the project. Although this was known to be far from ideal, the project timeline and resources did not allow for complete community mobilization and organization, given that activation and organization of the migrant households would have taken significant additional efforts and resources. Realizing these limitations, at later meetings residents, including migrants, from both sides of the village were included, such as in the gaming workshops (see below section 6.3).

**Negotiated Approach Task 2: Reaching agreement on process design**

In the course of the first year of engagement, farmers and fishermen groups were formed to represent the community in the project's negotiated approach process. The traditional livelihoods of these groups in the community were under pressure, among others as a result of increased selling of land to land developers and migrants.

These village negotiation groups were supported in a participatory problem analysis, for which the main steps had been outlined in a local Bengali guide for the negotiated approach. This guide had been developed by the local partner NGO JJS, after the
support received from international negotiated approach experts from the Netherlands and India (see Kolkata experiences). The local project partner JJS also facilitated the implementation of these steps with the community, to prepare them for a purposeful dialogue with other stakeholders, including government officials.

Contact with government officials had been initiated early on in the project. Although government resources seemed constrained, the rapport with government officials based in Khulna City, including the Khulna Development Authority and
district agencies, was good and no real problems were foreseen for later stages. The good relations with these government officials in Khulna benefited from the existing relations of local partner JJS, which was based in Khulna, and the nation-wide reputation of the research partner BUET in Dhaka.

## 6.2 Transdisciplinary Research Phase B: Co-creation of solution oriented knowledge

**Negotiated Approach Task 3: Joint fact-finding and situation analysis and Task 4: Solution analysis**

*Community level activities*

The village negotiation group conducted a problem analysis. This problem analysis used the results of the survey conducted in the village and it included a social map with water sources and water uses in the villages, a stakeholder mapping, and an identification of several water-related problems. Three priority problems were identified:

i. accessible safe drinking water,

ii. canal encroachment and water logging,

iii. waste dumping by the city corporation.

These priority problems followed community needs and priorities, and thus were not all directly related to groundwater. Nevertheless, all three issues were incorporated in the project, even if the research interface for some of these problems was weak or absent. This was part of the implicit process agreement between project team and local community stakeholders. For
these three priority issues, researchers contributed their expertise and the local team helped the villagers to develop a small-scale management plan to address them.

Although migrants were not represented in the smaller village negotiation group, they were part of the research activities and were invited at some of the workshops. This suggested that the drinking water problem was also acutely felt by this section of the community. The group of migrants included both relatively wealthy and relatively poor households. Most migrants in the
vulnerable category used one of the three shared tube wells in the village and they (as well as other poor households) needed over one hour to collect water. This was especially a problem for the women, who were responsible for water collection.

*Research contributions to situation and solution analysis*

The groundwater researcher made several field visits to the village for primary data collection. During these field visits,
awareness on groundwater issues was raised through discussions with village community members. When first results were available, information on groundwater quality and over-pumping fed into the village negotiated approach process, among others via a lecture by the researcher on groundwater scenarios to the village water group. Further, researchers assisted with a Bengali translation of key groundwater terminology.

Community-based groundwater monitoring was considered during the project mid-term deliberations as a way to combine
village capacity development with groundwater research data collection. Eventually, this was not initiated, mostly due to project timelines and research priorities – in which a PhD study was a key element, for which data collection results would come too late – and resource constraints.

Research findings from the developed groundwater models indicated that local groundwater abstractions might not have a very large effect on local groundwater availability, which seemed more influenced by regional level forces tied to the river (Hasan
et al., 2019). This provided a confirmation of the participatory management plans, reducing the need to focus on local water-demand management issues for the short-term.

The development of a community-based participatory approach for institutional analysis was a core objective for the institutional research component. This approach was developed with the Kolkata and Khulna sites in mind. The steps in the approach were mostly explored and applied with the Khulna village community for the prioritized drinking water issue. During
the earlier stages in the project, an institutions brief on water supply and groundwater management was prepared, translated and discussed with participants in the village. The brief outlined the different organizations, rights and responsibilities for water resource management in Bangladesh. It also contained an infographic about the process for tube-well applications in peri- urban areas. This was accompanied by a de-briefing workshop with the village negotiation group, other community members and some government officials, where they reflected on these institutional structures. This supported the village
group in its awareness of the situation, and the stakeholder mapping for the solutions planning. At the same time, the institutional research used local reports of the negotiated approach meetings as a source of data on the community's problem perceptions.

Combined, the groundwater and institutional research efforts helped to deepen the knowledge of villagers about the groundwater management situation, in such a way that they were able to talk about this to the authorities. Their increased
knowledge and their ability to use officially recognized terminology, empowered villagers in their communication with the government officials.

**6.3 Transdisciplinary Research Phase C: Re-integrating and applying produced knowledge in science and social practice**

**Negotiated Approach Task 5: Forging agreement and Task 6: Communication with constituencies**

For the direct engagement with the government officials, the members of the village negotiation group were trained by the local NGO (JJS) and at a local university in Khulna on advocacy and strategy development.

The community negotiation group shared their water related problems with the identified authorities and agencies during a workshop meeting. This workshop enabled the community negotiation group to continue discussions with the individual water related authorities after the meeting. During these individual follow-up meetings, there was more time and opportunity to

discuss the specific problems and the authorities shared their plans and initiatives for overcoming those problems. Through these follow-up meetings, all three priority problems were taken up by various government authorities. The public health agency in charge of rural water supply committed to test drillings for a functioning deep tube well for drinking water in the village, in recognition of the declining water tables and the need for sufficient safe public drinking water supply points. The Khulna City Corporation cleaned the waste dump near the village and selected two new sites for landfilling elsewhere. The

local level government administration (called upazilla) took the initiative to remove canal barriers. Linkages with the Bangladesh Water Development Board and an ongoing internationally funded water management project resulted in an effort to further clean up the drainage canal.

The issue of canal encroachment and water logging was caused by clogged drainage canal structures but was exacerbated by local fish farming practices. Although fish farming was decreasing, a few powerful local elites did engage in fish cultivation.

Branches of the drainage canal were captured for fish cultivation. However, the bamboo fences and temporary dykes for fish cultivation reduced the water flow and exacerbated problems with drainage during heavy rains and water logging. The fish cultivators earned a lot of money and shared the benefits with local powerful individuals. This made it difficult for the local open-water fishermen and smaller farmers to deal with them. The village negotiation group first tried to involve these powerful canal encroachers in the project meetings, but they were not interested as they thought they would lose their livelihood. After

these initial efforts, the focus was put on capacity strengthening of the more marginalized groups, to help them to negotiate and improve their knowledge. Illegal activities, especially canal encroachment, were condemned by the government officials at the meeting and in later press coverage.

*Applying produced knowledge on drinking water management and institutions*

The institutional research followed a sequenced design for a participatory analysis process, aligned with the negotiated approach activities in the Khulna village (Gomes, Hermans and Thissen, 2018). In the final stage, this resulted in two gaming simulation workshops, where the analytical results were shared and discussed with participants in a structured role-playing format. An important reason to opt for this format, rather than a formal report or presentation, was the low level of literacy in the peri-urban village community. In addition, gaming simulations are known to be effective means of communication, if well

designed and facilitated. One workshop was held with the village community and a second workshop was held with government representatives from different agencies involved in drinking water and/or groundwater management in Khulna. The purpose of these workshops was for participants to explore strategies to address drinking water related problems in peri-urban Khulna. The workshops provided a platform for research uptake where the results of the institutional analysis were shared with local stakeholders in the form of a role-play game. The workshops were valued by the community participants

with suggestions for future uses to engage more groups (Gomes, Hermans, Islam et al., 2018; Gomes, 2019).

**Negotiated Approach Task 7: Monitoring agreed actions and sustaining societal impacts**

At the end of the project period, peri-urban water issues were being discussed at different levels of government, at universities and in the local media. A gaming simulation seminar and workshop were organized at the local university, as well as more conventional workshops and meetings. A linkage between community and government stakeholders had been developed. A peri-urban water forum was established with representatives of several communities, beyond the project village community only, related government authorities and civil society. This forum connected the Shifting Grounds project with similar projects and activities in other peri-urban villages around Khulna City. In this way, the peri-urban water forum could become sustainable.

A small spin-off project after the ending of Shifting Grounds continued work with the approach for participatory institutional analysis, whereby local professionals were trained to develop gaming workshops for other water-related issues, with support from JJS and Delft-based researchers. Although this enabled a bit more monitoring after the project end, the longer-term monitoring in Khulna suffered from similar limitations as in Kolkata (see above).

**Negotiated Approach Task 8: Strengthening capacity of participants to become equal partners in negotiations**

Community empowerment for water management in Bangladesh carries a specific gender-challenge. As women were most affected by drinking water problems, they were interested in participating. During initial field visits it was observed that, though women got a voice in village matters, the last word was always with the men. In the community negotiation group that spoke with the government officials, three of the six members were women. In the first workshop, only the men spoke and when we asked women to speak, the men did not allow them. Towards the end of the project, the women had no problem to speak during workshops and meetings. They actively participated in the negotiation role-playing game and during the final project workshop the women eventually discussed directly with senior government officials.

The success on empowerment, fairness and legitimacy was mixed in the project. Although efforts were made, both powerful and powerless groups were eventually excluded from some of the most intensive part of stakeholder participation activities in the village. For the group of migrants, with its large heterogeneity, this was mostly dictated by limited timelines and resources. For some of the powerful local elites engaged in fish cultivation, their exclusion seemed a willing choice, possibly seeing the process as a potential threat to their business activities. Within the group that was represented, the role of women seemed to grow over time.

## 7. Discussion of the Shifting Grounds project experiences with transdisciplinarity and empowerment

The project experiences described for the research and negotiated approach activities in peri-urban villages near Kolkata and Khulna, partly confirm the challenges known for transdisciplinary research trajectories. Project designs had to be continuously adapted and changed, and, in some ways, had been over-ambitious. Project activities had to be tweaked to the site-specific conditions and constraints, and as a result the activities across countries were not uniform, neither for the stakeholder

empowerment, nor for the research components. The resulting process was very intensive and time-consuming, for all parties involved, much more than for comparable projects aimed either primarily at research, or primarily at direct local water management interventions. Nevertheless, there also seem to have been synergies and added values. The societal process with community and government stakeholders shaped research activities and results, and in turn these research activities and results influenced the societal dialogue within communities and between communities and government officials.

In addition to the confirmation of these prior experiences, the Shifting Grounds experiences also surfaced new challenges and responses, not previously emphasized in reviews for transdisciplinary research. These are summarized in Table 3. These challenges and responses are specific to transdisciplinary research in situations where power and empowerment shape the process of co-creation of knowledge and solutions and their implementation. These responses are context specific, they do not provide cure-alls and they come with their own dilemmas and limitations. Some of the responses in fact are about accepting limitations or looking for satisfactory rather than optimal solutions. For instance, even if our stakeholder mapping captured the presence of a significant number of migrant households in the Khulna peri-urban village early on, their heterogeneity and low level of organization, combined with our limited project resources, did not enable us to enable their effective representation in our project activities.

Challenges and responses similar to ours will be familiar to experts working on community or stakeholder empowerment projects, but so far, remained either invisible or fairly abstract for transdisciplinary research. Table 3 is a step in filling this lacuna, based on our experiences in this project.

INSERT TABLE 3 AROUND HERE

In our project, we have adopted a negotiated approach to deal explicitly with power in transdisciplinary research, with some practical lessons captured in Table 3. As we have explicitly engaged with the power-dimension, we have seen power structures and inequalities play out and affect our work. For instance: the contentious issue on the water bottling plant and the local level panchayat politics that proved more important than the state level government administration in Kolkata, the role of fish farming by local elites and the low-level of organization of migrants in Khulna, and the role of women in both locations. The responses in Table 3 illustrate that these power issues could not necessarily be solved, even if they could be observed. A full negotiated approach process will take more than a few years and may at times be more intensive than what we could facilitate in our transdisciplinary research project. Therefore, it is more accurate to talk about power and politics in transdisciplinary research in terms of *empowerment*, instead of in terms of solving power inequalities.

Empowerment is a dynamic process that may never fully end. We do have indications that we have been able to make a fruitful contribution to this empowerment process in our project villages, through our explicit engagement with power. These indications include an increased visibility of the heterogeneity within the peri-urban communities, the recognition of different groups such as women, youth and migrants, and their increased participation in, and knowledge of, local

groundwater management processes. At the same time, our longer term impacts remain unknown. This may be typical for transdisciplinary research, where research interventions are mixed with, and followed by, other activities that also influence the problems at hand. Attributing longer-term impacts to specific projects or activities constitutes a research challenge of its own.

**8. Conclusion**

We have applied a negotiated approach in transdisciplinary water research, to do justice to the importance of power dimensions and empowerment, instead of assuming a neutral co-learning process. Overall, our experiences confirm that, at least when working with relatively vulnerable and underrepresented local communities, employing a negotiated approach is useful, if not critical. It forces researchers to pay much more attention to the social and political realities, and to community leadership and
645 representation, early on in the process. Our experiences further confirm the earlier reports on transdisciplinarity that stress the importance of early and ongoing joint problem formulation, the importance of flexibility, and the struggle to match longer-term ambitions with short-term needs of both researchers and societal stakeholders.

In addition to these insights, we also added a specific list of challenges and responses for transdisciplinary research that seeks to actively engage with power and empowerment. This includes the use of careful observation, stakeholder analysis and social
surveys to map existing power structures, and the tuning of knowledge co-creation activities to those power structures with an eye for feasibility and longer-term risks and consequences. Also important is flexibility, including the flexibility to accommodate needs of societal actors that may be outside the core domains pre-identified for research contributions. This list, which resulted from our project reflections, will help to build a better articulated set of principles and guidelines for future transdisciplinary water research.

An uneasy conversation that we will need to engage with more, is one about the limits of transdisciplinarity and the various dilemmas it raises. Whereas many overviews result in an ever-expanding list of principles, tools and approaches for an ideal-type transdisciplinary process, reality is served better by a perspective on transdisciplinarity as yet another craft and "art of the feasible" in which tradeoffs between multiple and sometimes conflicting objectives and perspectives need to be made.

An overarching message for transdisciplinary water researchers, is to engage with power and politics more explicitly, as part
of this process. This is critical from the (pre-)inception phase of activities, as a key input for problem structuring and research agenda-setting. Even if some researchers will feel uneasy with this dimension, it cannot be ignored in transdisciplinary research. Ignoring power and empowerment, is just another way of dealing with it – allowing existing structures and forces of power and politics to co-shape transdisciplinary results in an unobserved manner. Engaging with power and politics is difficult but fundamental to societal change.

**Author contribution**

All authors participated in the design of the study, its execution, and reflection on the experiences as reported in this paper. After a joint reflection and writing workshop, LMH and VN took the lead in the further preparation of the manuscript, with all co-authors contributing other specific parts, comments and edits.

**Competing interests**

The authors declare that they have no conflict of interest.

**Acknowledgements**

We would like to thank the village communities, the government officials and the experts in the Kolkata and Khulna areas, who engaged with us on the learning trajectory reported here. Thanks are also due to the Shifting Grounds project advisory group members from India and Bangladesh, to Vijay Paranjpye and to all the colleagues and students who supported the Shifting Grounds project in various ways. We also benefited from the numerous discussions and interactions with the researchers, advisers and committee members in the NWO Urbanising Deltas of the World programme.

**Financial Support**

The research reported here was funded by the Urbanizing Deltas of the World programme of the Dutch Research Council NWO, under grant number W 07.69.104, project "Shifting Grounds: Institutional transformation, enhancing knowledge and capacity to manage groundwater security in peri-urban Ganges delta systems".

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

**List of Tables:**


**Table 1 Challenges and strategies in transdisciplinary water research (source: Lang et al., 2012; Steelman et al., 2015)**

| Phases and challenges | Exemplary strategies (Lang et al., 2012) | Coping strategies (Steelman et al., 2015) |
|---|---|---|
| **Phase A: Problem framing and team building** | | |
| Lack of problem awareness or insufficient problem framing | Primary study to build problem awareness | Iterative refinement of problem based on on-going discussions |
| Unbalanced problem ownership | Joint leadership | Hiring community-based monitors and research design with inputs of community members |
| Insufficient legitimacy of the team or actors involved | Stakeholder mapping, creating structures that enable participation | Continuous effort to broaden stakeholder representation as problem aspects are re-framed |
| **Phase B: Co- creation of solution- oriented transferable knowledge** | | |
| Conflicting methodological standards | Systematic comparison of methods, demonstration projects | Use of creative scientific publishing opportunities, more on process than on results |
| Lack of integration | Structured and formative knowledge integration methods | Identify publishable units that document smaller aspects of broader research effort, responsive to use to partners |
| Discontinuous participation | Design low thresholds for, and appropriate levels of, participation | Create reflexive experience and regular contact with local leaders |
| Vagueness and ambiguity of results | Specification and explicit conflict reconciliation | Collect more data to create greater confidence and delay conveying findings to broader community until realistic solutions can be recommended |
| Fear to fail | Initialize actions first to stimulate learning-by-doing | N/A (Did not apply) |
| **Phase C: Re- integrating and applying the produced knowledge in both scientific and societal practice** | | |
| Limited, case-specific solution options | Comparative studies for generalizable results | Continue to collect, scientific credibility data set will grow with time. |

| | | |
|---|---|---|
| Lack of legitimacy of transdisciplinary outcomes | Take into account existing socio-political context into design | Continue to build research-informed constituencies. Maintain long-term, on-the-ground presence |
| Capitalization on distorted research results | Establish ongoing collaborative and reflexive discourse | N/A (too early in process) |
| Tracking scientific and societal impacts | Employ advanced evaluation methodologies | N/A (too early in process) |

**Table 2. Activities for the negotiated approach (NA) and how they could help address some known challenges in transdisciplinary research (TDR) (Source for NA: Koudstaal and Paranjpye, 2011)**

| NA Tasks | Explanation of NA tasks | TDR challenges addressed by NA task |
|---|---|---|
| Task 1: Preparing the process | Understanding the past initiatives and existing social arrangements<br>Selecting committed participants that represent a 'balance of power'<br>Identifying the broad areas and boundaries of the intervention | Lack of problem awareness<br>Unbalanced problem ownership |
| Task 2: Reaching agreement on the process design | Understanding of the institutional context, its possibilities and limitations by all participants<br>Specifying the agenda and procedures while allowing flexibility | Insufficient legitimacy of the team or actors involved |
| Task 3: Joint fact-finding and situation analysis (problem analysis) | Ensuring that the participants understand each other:<br>Clarity on the backgrounds, aspirations and interests of various stakeholders<br>Collecting and understanding of objective information on the natural system<br>Joint fact-finding might be needed | Lack of problem awareness<br>Unbalanced problem ownership<br>Insufficient legitimacy of the team or actors involved<br>Discontinued participation |
| Task 4: Solutions analysis | Establishing a prior agreement on the criteria, separate from the weight given to them by different stakeholders<br>Considering and discussing all the solutions that are identified by the stakeholders | Discontinued participation<br>Limited solution options<br>Lack of legitimacy of TDR outcomes |
| Task 5: Forging agreement | Focusing on commonalities and using an iterative process of identifying, analysing and selecting solutions<br>Positional bargaining by one or more parties might require active mediation by an independent outside facilitator | Lack of legitimacy of TDR outcomes |
| Task 6: Communication with constituencies | Allowing the stakeholder representatives ample time and documented information to maintain the communication with their constituencies | Lack of legitimacy of TDR outcomes |

| | | |
|---|---|---|
| Task 7: Monitoring agreed actions | Making long-term commitment by the stakeholders for monitoring the implementation of agreed actions and the impacts of those actions | Tracking scientific and societal impacts |
| Task 8: Strengthening the capacity of participants | Extensive training of local communities to build the knowledge and skills they need to become equal partners in negotiations – among themselves and with the other key stakeholders and government officials | Unbalanced problem ownership<br>Lack of legimacy |

**Table 3. Challenges and responses for the negotiated approach (NA) tasks in transdisciplinary water research (TDR) in peri-urban cases**

| Phases and tasks | Observed challenged in relation to power and empowerment issues | Strategies used in response |
|---|---|---|
| **TDR Phase A: Problem framing and team-building** | | |
| NA Task 1: Preparing the process | The existing balance of power and socio-political dynamics could not be observed by the project team at the start of the process (there were neither time nor resources to conduct a thorough study prior to initiating the engagement in Kolkata) | Assure that the selected community is the "best available" project site, through careful selection process with selection criteria that include the community stakeholders' competence and willingness to engage |
| | | Pay continuous attention to socio-political dynamics and modify process designs when needed, throughout the duration of the project |
| | Differences in existing community organization structures caused uneven representation of groups in the negotiation process (Khulna and Kolkata) | Observe and accept an uneven representation in the negotiation process as a limitation of the project |
| | | Include the groups that are under-represented in research data collection and analysis to make interests and roles visible |
| | Large power distance existed between government decision making and communities (Kolkata) | Use the research process and the participation of an international science team as leverage to engage with government decision makers |
| NA Task 2: Reaching agreement on process design | The project team members were learning about (NA) process design and steps themselves | Ask help from an internationally recognized local NA expert for the external facilitation of the first workshops |
| **TDR Phase B: Co- creation of solution- oriented transferable knowledge** | | |
| NA Task 3: Joint fact- finding and situation analysis (problem analysis) | The competence to articulate and share problem views was different among the stakeholders and project team members alike | Use visual methods for problem appraisal (e.g. "social village maps") |
| | | Develop a joint language through the establishment of a shared vocabulary and a list of terminology |

| | | Use of role-play games to share the analysis insights |
|---|---|---|
| | | Use of low-cost community testing kits |
| NA Task 4: Solutions analysis | Urgent problems demanded short-term visible results for the community stakeholders (Khulna and Kolkata) – threatening their longer-term engagement in the TDR process | Free up project resources and mobilize additional resources to work on the emergent issues of immediate need in the villages, also if they were a less good fit for the research agenda of the project |
| **TDR Phase C: Re- integrating and applying the produced knowledge in both scientific and societal practice** | | |
| NA Task 5: Forging agreement | The deeper lying conflicts and issues could not be addressed within the project's limits | Focus on other significant issues for the community and in the research |
| | Some powerful actors did not engage (fully), making an agreement with them difficult | Ensure the participation of other actors with influence (local government actors mainly) |
| | | Mobilize media (Khulna) |
| NA Task 6: Communication with constituencies | Language barriers existed between (some of the) team members and communities; Illiteracy levels were high among the local community members | Prepare specific stakeholder communication materials using translations and visual images |
| | Heterogeneity was large in the peri-urban community groups | Organize small-scale community meetings with different sub-groups (frequently) |
| NA Task 7: Monitoring agreed actions | The limited project timespan, with first agreements reached only after the initial years, made longer-term monitoring by project team members difficult | Monitor within the project time frame through continued periodic visits and workshops with community and government representatives |
| | | Establish a platform linked to other projects and initiatives with continued monitoring by local project partners (Peri-urban water forum Khulna) |
| NA Task 8: Strengthening capacity of participants to become and remain equal partners in negotiations | Sustained capacity was threatened by short project timespans (& capacity strengthening challenges discussed with some of above tasks) | Link up to the existing structures for collective action and planning (Village Water Council Kolkata village) |
