# Peer review of "Power and Empowerment in Transdisciplinary Research: A Negotiated Approach for Peri-Urban Groundwater Problems in the Ganges Delta"

_Hydrology and Earth System Sciences, 2021_

## Author Response (AR1)

**Reply Note 2: Additional notes to editors and reviewers on revised version of manuscript**

*13 January 2022.*

*This first page has some minor deviations and addition to the earlier reply note that we shared on 3 December 2021 (added below on pages 2-5). We hope that the editor and reviewers can use both notes to aid their review of the revised manuscript.*

We have implemented most of the revisions as planned in the December response. Some minor changes to the announced revisions were made, as a result of seeing the text revisions take shape.

The Introduction has been revised in line with the reply note. It now starts with groundwater management problems in peri-urban areas, and the multi-actor complexities and power dimensions involved in those.

In this revised Introduction, also the relevant parts of earlier Section 3.1 have been used. These have not been repeated in Section 2, meaning that Section 2 does not start with a further expose on peri-urban water management challenges, but starts with TDR, as in the original manuscript.

Discussion of power issues in Sections 5 - 7: Power is discussed throughout. We've listed/summarized below:

- Section 5.1: Formal institutional hierarchy and power structures. Village leadership and politics (including powerful drinking water bottling plant owner).
- Section 5.2: Gender and youth participation. Changing status and income dynamics in peri-urban village. Illiteracy vs knowledge of legal rights as partner for dialogue with officials.
- Section 5.3: Empowerment via improved arsenic and health knowledge. Factual maps arsenic affected well locations to overcome reduced participation of political opposition supporters.
- Section 6.1: Migrants and representation, linked to community homogeneity and power of self-organization.
- Section 6.2: Diversity within migrants, access to shared wells for vulnerable households and women. Knowledge and terminology provided by researchers, as empowerment factors for villagers in their communication with government officials.
- Section 6.3: Role of powerful local elites in fish cultivation and water logging. Role-playing games for institutional insights with illiterate village group members. Gender issues in community empowerment. Inclusion and exclusion of village groups in the TDR process.
- Section 7: At the end of this Discussion Section, we have added a short summary/discussion of how we've dealt with issues of power and empowerment in the two project villages.

Minor corrections:

- Consistent spelling of "Negotiated Approach", "negotiated approach" or "NA": We have opted to write "negotiated approach" in full and without capital letters throughout the paper. Only in Tables or section headings do we use the abbreviation NA.
- We followed a similar approach for transdisciplinary research (abbreviated in tables and some section headings to TDR).

**Original Reply Note 1: Author Comments / Reply to Referees**

**Power and Empowerment in Transdisciplinary Research: A Negotiated Approach for Peri-Urban Groundwater Problems in the Ganges Delta.**

HESS-2021-419 (Reply date: 3 December 2021)

**Reply to Referee 1 (Roman Seidl)**

We would like to thank Dr. Seidl for his supportive and useful review. It is nice to read that he considers our manuscript worth publishing.

*Main suggestion*

Section 1: Introduction

The main suggestion is to start more clearly with the problem description, to clarify the 'why' of the paper. This resonates with comments made by Referee 2, and we agree. In the original submitted manuscript we started with transdisciplinary research (TDR), partly because of the special issue on TDR to which we hope to contribute. For a revised version, we would rework the Introduction to the following order, in line with what is suggested in the Review report:

1. Sustainable groundwater management in peri-urban areas is urgently needed but this challenge involves various actors, none of whom can do this alone. Groundwater challenges need to be approached both from an equity and sustainability perspective.
2. Actors have partly competing interests as well as power differences. Furthermore, limited information and knowledge make it hard to assess the consequences of different groundwater management strategies. (in other words: a "wicked" problem situation).
3. Such situations (wicked problems) are the types of problems that TDR hopes to support.
4. The 'power'-dimension is often lacking in TDR, but is essential in this problem (as it is for many other water-related problems). As highlighted by political ecology analyses around water governance, power is a key factor shaping differential access to resources.
5. We explore ways to resolve these groundwater challenges by combining TDR with a negotiated approach.

Section 2:

This changed Introduction, being more explicit about the challenge of peri-urban groundwater management, and tackling it via TDR and the negotiated approach, will then also mean that some changes in Section 2 are needed:

In a revised version, Section 2 would start with what is now in Section 3.1, to briefly outline the challenge with peri-urban groundwater management in South Asia. Then it would go into the details of TDR (currently section 2.1), to end with the contributions of the negotiated approach in this context (currently section 2.2). In this, we will seek to make what is currently section 2.1 more to the point, while we hope to add a bit more information on the negotiated approach (in what is now section 2.2, in response to Referee 2).

We will reconsider Table 2, probably to confine it to the negotiated approach only, to clarify differences and synergies between the two approaches. (Referee 2 made a similar point on Tables 1 and 2.)

*Further remarks:*

We agree with the further remarks and we think we can resolve them if allowed to do so by the editors. Specifically:

Page 4 line 126 – we will add a reference to companion modelling, with which we are familiar

Page 6, line 182 – Prisoners' dilemma: This came from the cited article. Given the role of this example from Gurgaon in the paper, we will probably remove the reference to the prisoner's dilemma here, rather than go in further details to explain this.

Page 7 and page 8 – we can add the requested explanations and elaboration for our project, as we agree these are important.

Page 9 and page 11: We will make sure to explain the background for the arsenic risk issues in the village in India more clearly, earlier on.

Some of the further remarks also are related to the main suggestion, which we hope to address in the way outlined above.

The smaller edits may not need a separate response here but will be processed.

**Reply to Referee 2 (Anonymous)**

We thank Referee 2 for a very helpful review. The points raised are valid and we hope that by addressing them, we can deliver on the potential that Referee 2 sees for a valuable contribution.

*Overarching concerns:*
*(four bullets raised, we follow the same sequence below)*

- *The 'why' of the paper is not clear*. Referee 1 made a similar point. We agree and propose to address this in the way that we outline in response to Referee 1, revising the Introduction of the paper. The suggestions by Referee 1 and Referee 2 for this are slightly different, but we expect that, starting with the challenge on groundwater management in peri-urban areas, the 'why' becomes clearer and more relevant for HESS readership.
- *Incorporate a greater focus on NA*. We will add more information on the negotiated approach (NA) in Section 2, and will see if we can be more to the point in the review of transdisciplinary research (TDR). This then may also help to give the NA more weight and visibility, as an 'equal' counterpart to TDR. We think that it may be better to align Table 2 exclusively with the NA, and not use it to compare/merge NA and TDR. Both referees indicate this was confusing.
- *More emphasis on power and empowerment is needed in Sections 3 – 6. Gender equality was mentioned but what about other power dynamics (literacy, influence, etc)*. We agree that this will help to sharpen the focus on power and empowerment. We have more information on these other power dynamics and can include this in a revised version in Sections 5 and 6.
- *Sections 3 to 6 could be better organized / structured.* We agree. Section 3 and 4 will have to be revised into a shorter Methodology section in Section 3 and a case intro in Section 4. The organization of especially Section 3 may further benefit from moving the background sub-section on periurban groundwater management (which is now section 3.1) to Section 2 (as a new section 2.1). This also fits with the new outline foreseen in response to Referee 1 comments. Sections 5 and 6 were structured to follow the main phases in the negotiated approach. We think that we can keep this structure, but we agree that we should make it clearer that this is the way these two sections are organized.

*Other comments:*
*(seven bullets raised, we follow the same sequence below)*

- Revise objective of the paper: We will put the case experiences that are the basis for our paper more clearly – and earlier – into Section 1. This would fit with the revised Introduction (see our response to Referee 1).
- Elaboration of TDR in context of human-water systems is lacking: Indeed we have not discussed this explicitly. Rather, in the discussion of TDR, we have often prioritized work related to water-human systems. We can make this clearer, separating the "water-TDR" more from the more generic works on TDR.
- Suggestion to add maps: In this paper, we only refer to the project sites in fairly general terms. We have taken care to report the essential information and to remove the sensitive parts from our case descriptions. Discussions of power and empowerment may have unforeseen future effects on local stakeholders. Therefore, we opted for a more anonymized case discussion. Part of this is not to add maps (even if we do have those). We realize that this is not very water-tight, and that a diligent researcher could probably still retrieve in which villages we have been working with our project team. Still, we believe it is the better choice for this manuscript. Hence, we prefer not to include maps.
- Add details on trust/confidence building: We can add more details as requested.

- Exclusion of migrants: We agree that acknowledging this is important. This is the reason that we have included this in our description of the process in Section 6. We can highlight this more, also in a more generalizable way in Section 7. It underscores the need for, but also practical limitations of, a stakeholder analysis for both NA and TDR. (Our stakeholder analysis at the start of the process, was done and did help us to identify the presence and importance of migrants in a significant part of the village. Still, eventually we could not include them fully in our project, for practical reasons and resource limitations. See also lines 405 – 410 of the original manuscript.)
- Task 1 in Table 3, assessment of competence and willingness: The competence and willingness of stakeholders were primarily assessed via site visits and discussions with community stakeholders early in the project. These visits started before actual site-selection, and conversations and visits continued to cover these aspects afterwards in early project stages. We can clarify this in the descriptions for Phase A in the two Sections 5 and 6.
- Task 7, problems with limited project timespan: We can elaborate on this in Sections 5 and 6, in the sub-sections for Phase C. (the problems that we see, are that the groundwater problems and negotiations cannot be resolved in 4-5 years' time. Ideally, the project is part of a longer-term process that continues, with maybe some external support from mediators and/or other researchers, well after 5-year project timelines. This need for longer-term process rather than 3-5-year projects is a limitation that is more often observed for TDR an NA processes alike).

*Grammar/Typos:*

Thank you for noting those. We will fix them.